# Criminalisation of clients: reproducing vulnerabilities for violence and poor health among street-based sex workers in Canada—a qualitative study

A Krüsi,[1] K Pacey,[2] L Bird,[3] C Taylor,[1] J Chettiar,[1,3] S Allan,[1] D Bennett,[2] J S Montaner,[1,4] T Kerr,[4,5] K Shannon[1,4]

▶ Prepublication history and additional material is available. To view please visit the journal (http://dx.doi.org/10.1136/bmjopen-2014-005191).

For numbered affiliations see end of article.

**Correspondence to**
Dr K Shannon;
gshi@cfenet.ubc.ca

## ABSTRACT

**Objectives:** To explore how criminalisation and policing of sex buyers (clients) rather than sex workers shapes sex workers' working conditions and sexual transactions including risk of violence and HIV/sexually transmitted infections (STIs).

**Design:** Qualitative and ethnographic study triangulated with sex work-related violence prevalence data and publicly available police statistics.

**Setting:** Vancouver, Canada, provides a unique opportunity to evaluate the impact of policies that criminalise clients as the local police department adopted a sex work enforcement policy in January 2013 that prioritises sex workers' safety over arrest, while continuing to target clients.

**Participants:** 26 cisgender and 5 transgender women who were street-based sex workers (n=31) participated in semistructured interviews about their working conditions. All had exchanged sex for money in the previous 30 days in Vancouver.

**Outcome measures:** Thematic analysis of interview transcripts and ethnographic field notes focused on how police enforcement of clients shaped sex workers' working conditions and sexual transactions, including risk of violence and HIV/STIs, over an 11-month period postpolicy implementation (January–November 2013).

**Results:** Sex workers' narratives and ethnographic observations indicated that while police sustained a high level of visibility, they eased charging or arresting sex workers and showed increased concern for their safety. However, participants' accounts and police statistics indicated continued police enforcement of clients. This profoundly impacted the safety strategies sex workers employed. Sex workers continued to mistrust police, had to rush screening clients and were displaced to outlying areas with increased risks of violence, including being forced to engage in unprotected sex.

**Conclusions:** These findings suggest that criminalisation and policing strategies that target clients reproduce the harms created by the criminalisation of sex work, in particular, vulnerability to violence and HIV/STIs. The current findings support decriminalisation of sex work to ensure work conditions that support the health and safety of sex workers in Canada and globally.

## Strengths and limitations of this study

- This is the first empirical study of how criminalisation and policing of sex buyers shapes sex workers' risks for violence and poor health, outside of a small body of research in Scandinavia.
- This study draws on data of the lived experiences of sex workers from 31 qualitative interviews triangulated with ethnographic observations, sex work-related violence prevalence data and publicly available police statistics.
- The data represented in this study reflect the experiences of primarily street-based sex workers, disproportionately impacted by policing and thus, may not be representative of other segments of the sex industry.
- This study is a qualitative evaluation post-implementation of a new enforcement policy targeting sex buyers rather than sex workers, and as such, likely underestimates the full impact of any legislative change to criminalise the purchasing of sex.

## INTRODUCTION

> Harassing the clients is exactly the same as harassing the women. You harass the clients and you are in exactly the same spot you were before. I'm staying on the streets. I'm in jeopardy of getting raped, hurt.
> —Jasmine, cisgender woman sex worker

There is now a well-established body of epidemiological and social science research globally pointing to the negative impact of legislation and policies that criminalise sex work on violence and other health risks including HIV/sexually transmitted infection (STI) among sex workers.[1–7] The criminalisation of some or all aspects of prostitution remains the dominant legal approach globally,[1 4] despite growing empirical evidence and clear international guidelines by the

WHO, UNAIDS, UNDP and UNFPA calling for full decriminalisation of sex work as necessary to promoting the health and human rights of sex workers (see online supplementary materials for a summary of legal and regulatory responses to sex work).

Enforcement-based approaches and policing within criminalised frameworks have consistently been linked to elevated risks for violence, and reduced ability to negotiate safer sex transactions, including prevention of HIV and other STIs.[1–3 6 8] Previous research indicates that in criminalised settings, policing strategies can range from surveillance and crackdowns to arrests or threats of arrest, intimidation by police, and police violence can be frequent and can go largely unreported.[2 9–12] These risks are amplified for the most marginalised and visible sex workers, those living in poverty and working on the street.[10 13 14] In an effort to avoid police, sex workers often move to outlying secluded areas to meet and service clients where there are few to no protections from violence and abuse, and reduced ability to refuse unwanted clients or services, including client demands for sex without a condom.[9 10 15–17] Criminalisation and policing force sex workers to rush or forgo screening prospective clients or negotiating the terms of sexual transactions before entering a vehicle, placing sex workers at increased risk of physical violence, rape and HIV/STIs.[9 18 19] Criminalisation of sex work also impedes access to safer indoor work environments, particularly for those most socially and economically marginalised. Previous research undertaken in Canada and internationally has indicated that indoor sex work environments with structural supports, including supportive management policies, security measures and access to HIV/STIs prevention resources, increase sex workers' control over sexual transactions with clients, including protections from violence, abuse and HIV/STIs, and promote sex workers' ability to access police protections in cases of violence.[17 20 21]

The criminalisation of sex work has also been found to prevent sex workers from reporting violent perpetrators and seeking legal recourse after physical or sexual assault.[9 22] There is also growing evidence that legislation criminalising sex work constitutes a significant barrier to accessing healthcare services, including primary care, HIV treatment and prevention and sexual health services.[16 23 24] Additionally, stigma and discrimination against sex workers are significantly amplified in settings where sex work is criminalised, and further reduce sex workers' ability to access police protections or health and social support services.[23 25]

### Demand criminalisation

Over the past decade, there has been increased interest by a number of higher income countries to attempt to eradicate prostitution through 'demand criminalisation', which criminalises the purchase, but not the selling, of sexual services. Sweden, Norway, Iceland and most recently France opted for 'demand criminalisation', despite the lack of evidence supporting this legal framework. Similarly, the European Union has also recently voted in favour of implementing this approach.[26] 'Demand criminalisation' was first implemented in Sweden in 1999. The primary objective of the law is to eradicate prostitution by eliminating demand. However, evidence from Sweden indicates that the law has been unsuccessful in meeting this objective.[27] Instead, a number of unintended consequences have been reported—namely, that it drives sex workers and clients underground to more clandestine locales, and is difficult to enforce due to the unwillingness of sex workers to testify against their clients.[28 29]

### The Canadian context

Canada is at a critical time in the evolution of its legal response to sex work. In December 2013, Canada's highest court, the Supreme Court of Canada, unanimously struck down Canada's core prostitution laws, deeming them unconstitutional for violating sex workers' constitutional rights, including the ability to protect themselves from violence, abuse and HIV/STIs.[30] The three laws struck down included prohibitions on communicating in public for the purpose of prostitution (sex workers or clients), operating a bawdy house and living off the avails of prostitution. Similar to the UK and other commonwealth countries, the Canadian Criminal Code has never criminalised the buying or selling of sex per se, however the laws prohibit virtually every other aspect of sex work, making it effectively impossible to engage in sex work legally. The hypocrisy of the criminalised prostitution laws and the unintended harms on sex workers' safety, health and human rights were a critical reason for the decision of Supreme Court of Canada.[30] The Court suspended the decision for 1 year to provide the Canadian government time to respond by either removing the struck down laws from the Criminal Code (and thus decriminalising sex work), or by bringing the laws governing prostitution in line with the decision. Following the decision, the Canadian government has moved quickly towards calling for criminalising the purchase of sex,[31] despite a lack of evidence that this change would address the harms associated with criminalised and quasi-criminalised approaches of regulating prostitution.

### New sex work enforcement policy in Vancouver, Canada

Vancouver, Canada, provides a unique opportunity to evaluate the potential impact of laws and policies that criminalise sex buyers (clients). In January 2013, the Vancouver Police Department (VPD) officially adopted new sex work enforcement guidelines that shifted the focus in policing sex work away from arresting or pressing charges against sex workers. These sex work enforcement guidelines emerged after two decades of tremendous violence and murder of street-based sex workers and strong pressure from sex work and other community organisations, legal experts and academic

researchers calling for reforms to local policing practices to better protect, rather than increase harm, isolation and risk of violence to sex workers.[32] The sex work policing guidelines set out a strategy to 'open communication' with sex workers and prevent violence against sex workers through prioritising their safety over enforcement measures such as arrest.[33] However, the policing guidelines did not address changes in enforcement of clients. In fact, on their website the VPD confirms that they continue to "target both pimps and customers, in locations where the impact of the sex trade has become unacceptable."[34] The result is that as of January 2013, Vancouver has been a de facto 'demand criminalisation' environment.

Therefore, the objectives of this study are to evaluate how a new local enforcement strategy that targets clients, but not sex workers, shapes sex workers' interactions with police and negotiation of their working conditions and sexual transactions with clients, with a particular focus on protections from violence and HIV/STIs.

## METHODS

This study is situated within a larger National Institutes of Health (NIH)-funded longitudinal qualitative and ethnographic research project investigating the structural and physical, social and policy features of work environments shaping sex workers' sexual health, violence and access to care in Vancouver, Canada. The research builds on longstanding partnerships and a community advisory board with sex worker, community, policy and health stakeholders since 2004, and runs alongside a sister project known as AESHA (An Evaluation of Sex Workers' Health Access). The AESHA cohort is a community-based longitudinal study of over 800 sex workers with biannual follow-up, focused on evaluating the physical, social and policy environments shaping sexual health, violence, HIV vulnerability and access to care among sex workers.[35] The research and outreach team include experiential and non-experiential staff.

This study draws on ethnographic fieldwork of street-based sex work scenes and qualitative semistructured interviews with street-involved sex workers about their working conditions, interactions with police and negotiations of health and safety with clients in the city of Vancouver, Canada, over 11 months (January–November 2013), following the implementation of the new sex work enforcement guidelines by the VPD in January 2013. Qualitative and ethnographic data were triangulated with sex work-related violence prevalence data from the AESHA cohort prepolicy and postpolicy implementation and publicly available police statistics that report on prostitution-related criminal code offences in the city of Vancouver.[36]

The lead author (AK) and coauthor (JC) conducted more than 40 h of ethnographic observation within known street-based sex work strolls in the city of Vancouver to assess level of police presence; shifts in working areas and police, sex worker and client interactions. All ethnographic observations were conducted within the context of regular weekly AESHA outreach shifts, which included provision of harm reduction supplies, food and referrals to social and health supports. Observation sessions lasted 3–5 h and took place during peak hours of sex work activity between 22:00 and 3:00. AK and JC recorded brief fieldnotes in a research log during the observation sessions and elaborated on them after each observation outing.

Interview participants were recruited through purposive sampling from the longitudinal cohort (AESHA), and aimed to reflect variation in demographics (eg, age, ethnicity and gender) and work environments (eg, geographic neighbourhoods, variation in street and off-street solicitation and transaction spaces). Eligibility criteria for the in-depth interviews included: (1) current sex work defined as exchanged sex for money in the past 30 days in the city of Vancouver; (2) identifying as cisgender or transgender woman and (3) aged 18 years or older. While the larger qualitative and ethnographic research and AESHA projects focus on a diversity of street and off-street (eg, indoor and online) sex work environments, this specific study aimed to examine the experiences of street-involved sex workers (ie, those soliciting and/or servicing on the street) given substantial data that criminalisation and enforcement disproportionately target this segment of the sex industry.[10 13 14] It should be noted that even within this context, many street-involved sex workers worked in street and off-street venues, including online and indoor informal and formal venues (see results).

The 31 semistructured interviews were conducted by two experienced qualitative interviewers (CT and AK) and facilitated by an interview guide encouraging broad discussions of working conditions, police presence and interactions and negotiation of health and safety in transactions with clients, post-VPD policy implementation (January 2013). The interview guide was developed based on existing knowledge of the research team and in collaboration with our sex worker, community and policy partners. We conducted all interviews at one of the two field offices in the city of Vancouver. Interviews lasted between 45 and 90 min, were audiorecorded, transcribed verbatim and checked for accuracy. All participants provided informed consent and were remunerated with a $C30 honorarium for their time, expertise and travel.

Interview transcripts and ethnographic data were analysed using thematic analyses to examine sex workers' interactions with police and negotiation of their working conditions and sexual transactions with clients, including protections from violence, abuse and HIV/STIs, postpolicy implementation. All textual data were analysed using an inductive and iterative process facilitated by the qualitative analysis software ATLAS.TI V.7. The initial coding framework was based on key themes reflected in the interview guide, participants' accounts and fieldnotes.

More conceptually driven substantive codes (eg, trust, discrimination, control over sex work transactions) were then applied. Verbatim narratives are reported using pseudonyms assumed by sex workers to ensure anonymity. Longitudinal quantitative data on prevalence of workplace physical and sexual violence among sex workers from the AESHA cohort were analysed by two 8-month time periods (prepolicy, 1 May–31 December 2012 vs postpolicy implementation, 1 January–31 August 2013). Analysis was conducted using SAS statistical software V.9.2 and restricted to sex workers in AESHA who solicited and/or serviced clients on the street. Descriptive frequencies and bivariate analyses were analysed to test for statistical significance by time period of VPD policy implementation (prepolicy vs postpolicy) and reported using ORs, 95% CIs and p values.

## RESULTS
### Sample characteristics
The sample for semistructured interviews included 26 cisgender and 5 transgender women who were sex workers (total n=31). The mean age of participants was 38 years (range 24–53). Overall, 21 identified as Caucasian, 8 were of Aboriginal ancestry and 2 participants were of other visible minorities. All participants had experience with street solicitation. The majority (77%, n=24) reported street solicitation as their primary way of connecting with clients, while others (23%, n=7) primarily used phone/text solicitation to connect with clients. Just over half (55%, n=17) primarily serviced clients in vehicles or outdoor public spaces, while 45% (n=14) primarily serviced clients in informal indoor venues (eg, hotels, client's place or their home).

### Sex workers' experiences with new sex work enforcement policy
Sex workers' narratives and ethnographic observations indicated that while police sustained a high level of visibility, they eased charging or arresting sex workers and showed increased concern for their safety. Most sex workers experienced a gradual change in policing over a number of years rather than an abrupt change in policing with the publication of new sex work enforcement policy in January 2013. Some women felt that they were interacting less with police as long as they solicited clients in two separate areas of the city that function as de facto sex work tolerance zones.

> There's more [police] presence. There's not very many more interactions. Before the interactions were always there. They'd come and pull you right outta car and, like push you over. They'd get away with it back then. You know. But now it's like they don't, interact.
> —Anna, transgender woman sex worker

The vast majority of participants, regardless of gender, ethnicity and primary place of solicitation, reported that their interactions with police when soliciting sex work clients are more positive and generally focus on their safety.

> Every time they pull you over it's strictly to ask you how you're doing, how things are. If there's any bad dates you want to report.
> —Fiona, cisgender woman sex worker

While participants in this study viewed this change as positive, the continued police enforcement of clients severely limited any positive impact of this change on their overall working conditions, risks for violence, abuse or negotiation of sexual risk reduction with clients.

### Continued police enforcement of sex buyers (clients)
Sex workers' narratives indicated that while police tolerated sex work-related activities in two separate de facto sex work zones, clients continued to be at risk of police scrutiny. Indeed, according to official police statistics sex work-related criminal code offences rose from an all-time low of 47 in 2012 to 71 in 2013 (see figure 1).[36] This represents a 51% increase in prostitution-related offences since the announcement of the VPD sex work enforcement guidelines in January 2013.

Unfortunately, no statistics are available regarding the proportion of sex workers versus clients who were charged. However, participants' accounts indicate that the 51% rise in prostitution-related offences likely reflects an increase of enforcement efforts targeting clients, as the majority of sex workers felt that their clients were currently the main targets of police (consistent with messaging on the VPD website).[34]

> I think the Johns or clients that I have probably worry the most about police.
> —Melissa, cisgender woman sex worker

> [Clients] do get stopped. A couple of my regulars they've been chased off the street, they're not allowed to come downtown anymore. Down here they're bad for that. I guess, it's more so not the girls they go after—it's the guys.
> —Maria, cisgender woman sex worker

Most sex workers reported that clients are at risk of being pulled over by police even before actually

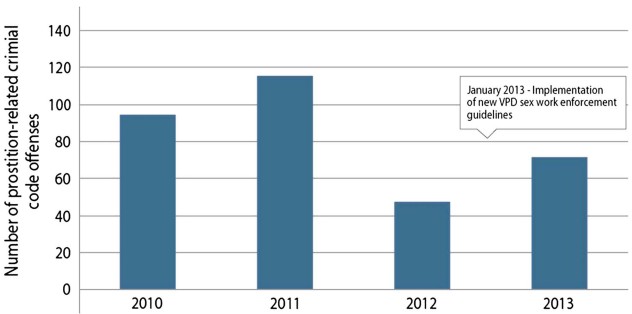

**Figure 1** Prostitution-related criminal code offences Vancouver 2010–2013 (Vancouver Police Department).

negotiating a sex work transaction by circling around known sex work areas. However, sex workers' narratives indicated that the riskiest time for attracting police attention was the moment when women entered the vehicle of a prospective client. In some cases, police used non-prostitution-related charges (eg, traffic violations) as a means to pull over the vehicle and target clients. Women expressed that while they used to be the main target of police, the recent experiences generally ended with the police allowing sex workers to leave, often without checking for outstanding warrants. Clients were being detained by police, and issued either a warning or a fine.

> Right now when they [police] pull you over they let you, the girl go, keep the guy. Before they would separate you and then make a big investigation. Try to catch you in lies or intimidate you and make you nervous and try to say you can get more charges.
> —Ruth, transgender woman sex worker

> One of my regulars, they gave him a five-hundred dollar disturbing the peace ticket. And they didn't run my name through. They just come out of nowhere, right? [Yeah, was it just for you getting in the car?] Yeah, I guess that was just the punishment, you know what I mean? So, I saw the guy a few times afterwards and he was saying he was going to fight and wanted me to be a witness and then I guess he thought afterwards, geez no. He told me he paid it online, I guess he has a family and that and he just wanted it to go away.
> —Jessica, cisgender woman sex worker

## Criminalisation of clients: limited effect on deterrence of sex work

Sex workers reported that when police target clients, some clients are deterred from purchasing sex on the street.

> No one will pull over if there's a car, a police car near you. It's, like, if they see the lights they'll disappear. You can see the difference in traffic. They're just gone.
> —Selina, transgender woman sex worker

Some sex workers, however, felt that rather than preventing clients from purchasing sex, police presence resulted in potential clients seeking out sex workers in a different area of the city.

> Once the guy that's looking for a woman sees a cop, in the neighbourhood, he's scared. So he'll go to another neighbourhood and find another woman somewhere.
> —Rebecca, cisgender woman sex worker

For participants in this study, the reality of living in poverty and marginalisation often combined with illicit substance use meant that even when police target clients, sex workers report that they continue to work for the obvious reason of earning an income. Ethnographic observation and sex workers' narratives indicated that

police enforcement of clients had no effect on deterring women from engaging in street-based sex work (box 1). Indeed, for many participants the enforcement of clients forced them to spend longer hours on the street to earn an income. Thus, contrary to the objectives of criminalising clients, impeding sex workers' ability to engage with potential clients did not result in less street-based sex work for these women. Instead, having access to fewer clients meant it was harder to earn an income and forced sex workers to accept clients or services (eg, sex without a condom) that they would otherwise reject due to safety concerns; this directly increased risks for physical and sexual violence and poor health, including HIV/STIs.

## Criminalisation of clients: severely limits sex workers' safety strategies

Our findings indicate that criminalisation and policing strategies that target clients reproduce the harms created by broader criminalisation of sex work. Analyses of prevalence of workplace physical and sexual violence against street-based sex workers in Vancouver indicated no statistically significant change in violence rates following policy implementation (OR=1.05, 95% CI 0.70 to 1.58; p=0.804). Specifically, in the 8-month period post-policy implementation, 24.6% (58/236) of sex workers experienced work-related physical and sexual violence

---

**Box 1** Criminalisation of clients: limited effect on deterrence of sex work

**Sex worker voices:**

While they're going around chasing johns away from pulling up beside you, I have to stay out for longer […] Whereas if we weren't harassed we would be able to be more choosy as to where we get in, who we get in with you know what I mean? Because of being so cold and being harassed I got into a car where I normally wouldn't have. The guy didn't look at my face right away. And I just hopped in cause I was cold and tired of standing out there. And you know, he put something to my throat. And I had to do it for nothing. Whereas I woulda made sure he looked at me, if I hadn't been waiting out there so long.
—Violet, cisgender woman sex worker

It pisses me off that they [the police] are there because basically what it comes down to is the shortest time that I'm out there, the shorter I'm on the street and the better I'm paid. But you [police] stand out there and you fuck up my business and scare away my dates. The longer I'm out there my chances of getting sick, raped, robbed, beat up whatever are greater so.
—Lisa, cisgender woman sex worker

Of course, 'cause no one's [clients] going to stop with them there. I'm not going to go home. So they're [police] not really doing anything, they're just keeping me out there longer. Really, if they would just leave me alone, I'd get a date and go home and they wouldn't see me. But that way I end up staying out there for hours 'cause I'm not going home empty-handed so I don't know what they think they're really achieving.
—Charlene, cisgender woman sex worker

 

(as compared with 23.7% (65/275) interviewed in the 8 months prepolicy in 2012), of whom 22% reported physical abuse and 14% had been raped postpolicy implementation (compared with 19.3% and 15.6% prepolicy, respectively).

Qualitative analysis of the sex workers' narratives reveal three key mechanisms by which criminalisation and police targeting of clients continued to severely impact sex workers' ability to negotiate their working conditions and transactions with clients, including protections from violence and HIV/STIs.

### Inability to screen clients and negotiate terms of sexual transactions

Sex workers' narratives emphasised that in the context where clients continue to be police targets, it remains in the clients' and sex workers' interest not to get pulled over by police. Therefore, sex workers continue to be forced to severely limit or forego screening of prospective clients or negotiating the terms of sex work transactions (eg, fee, sexual services and condom use) before getting into a vehicle (box 2). In addition to entering a prospective client's car swiftly, participants reported that in order to reduce the risk of attracting police attention, potential clients point them to an alleyway away from the main street with limited lighting to allow them to enter the vehicle undetected by police (box 2). Policing of clients thus directly undermines sex workers' ability to

---

**Box 2** Criminalisation of clients: severely limits sex workers' safety strategies

**Sex worker voices:**
Well, usually I try to hop in the car right away, right? 'Cause I don't want to get seen talking, in case a cop drives by or something. […] I'll hop in and then we can like negotiate and talk, you know? First I like to make sure that nobody's around or following or anything.
—Maria, cisgender woman sex worker

To avoid police they [clients] drive by couple times and they point. They point at like a place where nobody's driving by. So they point and that means to go follow them with the vehicle and then they'll stop […] They go somewhere different in an alley or something. They just leave like the window open and then you just, get in. [But would you talk to them first?] Um no well when they're trying to avoid police like that you just get into the vehicle, right.
—Jane, transgender woman sex worker

Sometimes the guy will drive up and just sort of wave or point to go down the alley or something like that somewhere else were he can pick me up. [How does that affect your safety?] You never know who it is right? And you can't really see his face, can't really see anything they could have a gun in their hand or. You know what I mean they could be a little bit drunk or something if you can't really see them very clearly, you know? And you don't you can't say hi or whatever before you get in. You have to just hurry up before the cops come.
—Laura, cisgender woman sex worker

---

*screen potential clients* including checking 'bad date' sheets for past violent perpetrators, detecting possible weapons or intoxication; and *negotiating the terms of the sexual transactions*, including where the date will take place, the fee and types of sexual services and use of condoms, before entering a vehicle. These practices of screening and negotiating the terms of transactions have been well documented as critical to sex workers' ability to control their health and safety, including protections from violence, abuse and HIV/STIs.[15 17 19]

Prior to the VPD policy, police frequently engaged in undercover operations in order to target sex workers and their clients. This practice continues to shape sex work transactions and, in the context of sustained criminalisation of clients, undercover operations negatively affect the ability of sex workers to screen their clients or negotiate the terms of sexual transactions. Participants' accounts indicated that rather than trying to assess the safety of entering a vehicle of a prospective client and negotiating the terms of the transaction, the initial interaction with a client is dictated by determining whether the sex worker is an undercover police officer. This usually involves the client and sex worker touching each other, due to the belief that undercover police officers are not allowed to engage in bodily contact without identifying themselves as police.

> Normally when you get picked up, you go: Are you a cop? No, are you? Nope. Prove it. And you, touch each other just to make sure, right? 'Cause cops can't do that. So that's the rule, if you're undercover you can't touch someone. Normally, a guy'll touch my boob, I'll touch his crotch. Or he'll touch my crotch, I'll touch his, right? That's just to verify okay, you're not a cop, right?
> —Martha, cisgender woman sex worker

### Displacement to isolated areas

Sex workers' accounts further indicated that in the context of continued criminalisation of clients, many clients demand on engaging in sex work transactions away from known sex work areas where there is heightened police presence. Participants reported that being alone with clients in often unknown, secluded, industrial areas where there is little chance for help puts women at increased risk of violence and rape and reduces their ability to negotiate the transaction on their terms, elevating their risks for client condom refusal and thus HIV/STIs (box 3).

### Inability to access police protections

The main objective of the police enforcement guidelines is to foster more trusting relationships between sex workers and police and prioritise the safety of sex workers in any police interactions. A striking feature of many sex workers' accounts was that police inquiring about their safety was perceived as a nuisance at best, and a form of police harassment at worst. In a context where clients continue to be police enforcement targets,

---

**Box 3** Criminalisation of clients: displaces street-based sex work to isolated areas

**Sex worker voices:**
Clients worry [about police]. Like for me I don't like going outside the neighborhood, right. Cause, you know what about if the guy turns out to be an asshole. […] That's how I do loose dates by not going where I'm supposed to cause they're afraid of cops. [So do you turn dates down sometimes?] Yeah sometimes but not all the time cause when I'm I'm really in need of money I will maybe try and go. But then I just try and get a good feel of them first.
—Jane, transgender woman sex worker

Clients are worried about police. To avoid police they wanna move to a different area. I don't want to go out of my zone right. […] Once you get out there, like you know their turf so it's harder for me cause it's their comfort zone so they act differently, you know what I mean. Yeah it never ends up good.
—Sandra, cisgender woman sex worker

We try to get away from the area as quick as possible. You know. So that we're not in the area. Right. The farther away you get from [name of sex work stroll], the better it is. You're not gonna get pulled over right? I'm just a little nervous as it's so quiet down there by [industrial area].
—Violet, cisgender woman sex worker

---

**Box 4** Criminalisation of clients: prevents access to police protections

**Sex worker voices:**
*Destabilising effect of police safety checks*
It's a drag, you know? I'm out there to make money, not waste twenty minutes talking to them [police]. And then I'm talking to them and half the dates that see me talking to them now think maybe I'm a cop, so they don't wanna stop, now they know the cops are around, they don't wanna stop, or they wonder what I've done to attract the cops so they don't wanna stop, it's just a hassle, you know?
—Charlene, cisgender woman sex worker

If the clients see you talking to the cops then they don't pick you up. [When police talk to me] they're respectful but they know that they're wasting my time so. They can do whatever they want. They're fucking up everything.
—Selina, transgender woman sex worker

*Reporting violence to police*
No I would never go to the cops [to report violence]. Because it makes it look like, we shouldn't be out there like we can't take care of ourselves. I feel like if I went and reported some of these things that it might do more harm to the working profession than do good. So I don't do that. Basically we have to fend for ourselves. They don't really like us to begin with.
—Rose, cisgender woman sex worker

I've needed the police's help with bad dates and they've done absolutely nothing. The fact that it's not legalized you kinda can't do it, you know.
—Charlene, cisgender woman sex worker

---

sex workers' narratives indicated it is difficult for police to fulfil their stated objective of prioritising the safety of sex workers. Even conversations between sex workers and police can have a destabilising effect, as any police interactions may scare away clients and have the potential to raise suspicion that a sex worker might be an undercover police officer (box 4).

Similarly, an important aspect of sex workers' safety is the ability to report theft, violence and sexual harassment to police. Currently, however, the majority of sex workers voiced reservations about reporting such incidents to police. Many sex workers, drawing on historic discrimination and maltreatment by police, doubted that police would take their complaints seriously and voiced that the continued criminalisation of clients constituted a significant barrier to reporting violence to police as any information about where they work could be used to refine enforcement strategies targeting clients.

## DISCUSSION

These findings suggest that criminalisation and policing strategies that target clients reproduce the harms created by the criminalisation of sex work, in particular, risks for violence and abuse. Contrary to the goal of criminalising clients, our findings suggest that this approach has limited to no effect on preventing street-based sex work and did not reduce the prevalence of sex work-related violence. Our analysis provides strong empirical evidence of the lived experience of sex workers indicating that the continued criminalisation and policing of clients, even when sex workers are no longer police targets, profoundly impacted sex workers'

ability to negotiate their working conditions and health and safety. Collectively, this research suggests that displacement to isolated areas and inability to screen clients or negotiate the terms of sexual transactions directly reduced sex workers' ability to refuse unwanted clients or services (eg, sex without a condom), thereby increasing risks for physical and sexual violence and HIV/STIs. In addition, despite improved relations between sex workers and police, continued police targeting of clients created mistrust of police and severely limited sex workers' ability to access police protections.

In a legal environment where clients remain the target of enforcement, our findings indicate a shared interest of sex workers and clients to remain undetected by police, forcing sex workers to rush or completely forgo client screening, and pushing sex workers to work in secluded areas away from street lighting and other passersby. These findings reflect earlier epidemiological and social science research that have consistently demonstrated a direct correlation between criminalisation and policing of street-based sex work and elevated odds of physical violence and rape, as well as HIV/STIs through client condom refusal.[5][6][10][11][15][18] This work has identified screening of prospective clients as essential to allowing sex workers to take safety precautions, including agreement on where the transaction will take place, checking 'bad date' reports describing the

personal characteristics and/or vehicle of known violent perpetrators and checking for the presence of weapons and intoxication.[12] In addition to screening, negotiating the terms of sexual transactions with clients before entering the vehicle, including the fee and types of sexual services, remains critical for sex workers to negotiate the terms of their work, and avoid risky sexual encounters (eg, unprotected sex). Without the opportunity to screen clients or safely negotiate the terms of sexual services, research has shown that sex workers face increased risks of violence, abuse and HIV/STIs.[15 17 18 37 38] Our findings resonate with evaluations from Sweden that reported that since the implementation of the law that criminalises clients, there had been an increase in violence experienced by sex workers, which was linked to greater risk taking in client selection due to the necessity of rushed negotiation with potential clients.[28 29]

Police undercover operations further reduce the safety of sex workers as initial interactions with clients focus on determining that sex workers are not undercover police officers, rather than allowing time to negotiate the details of the transaction, including condom use, type of service and price. The negative impact of undercover police operations on the safety and health of sex workers is of note as it is one of the main enforcement strategies available to police in a context where clients are criminalised. Sex workers also described how police used other non-prostitution-related offences (eg, administrative laws such as traffic violations, public nuisance) to target sex workers and clients.

The results of this study further highlight that in a context where sex buyers are criminalised, sex workers continue to be displaced, as many clients insist on engaging in sex work transactions away from police scrutiny. Being displaced to unknown, secluded, industrial areas where there is little chance of receiving help when needed is linked to increased risk of violence and rape and reduces sex workers' ability to negotiate the transaction on their terms, including condom use.[15 17 37 38]

Evidence from Canada and globally has consistently shown that criminalisation of sex work prevents sex workers from accessing police protection, whereby police become adversaries as opposed to safety mechanisms.[6 9 10 22 29] A clear example is the case of the detrimentally flawed police investigation of the serial murder of the missing women in Vancouver, Canada; where criminalisation and historic discrimination by police were found to be key factors in putting police in an adversarial relationship with sex workers.[32] In addition to the deeply engrained stigma of sex work and the historic police discrimination of sex workers, our findings indicate that when sex work clients are police enforcement targets, many sex workers remain reluctant to seek police protection. Sex workers worry about disclosing details of how they operate and where they work for fear police may use this information to refine their enforcement strategies that target sex work clients. This study

also identified that in a context of continued police enforcement of sex work clients, even conversations with police about safety can have a destabilising effect, as any police interactions scare away clients and have the potential to raise suspicion that a sex worker might be an undercover police officer or a police informant. This policing practice deters women from interacting with police and undermines the main objective of the VPD policing guidelines, which is to prioritise the safety of sex workers. In addition to scaring away potential clients, and thus potentially reducing the income women rely on, being labelled as a police informant can place a woman at the bottom of street hierarchy and may place her at severe risk for violence.

While rhetorically powerful and politically appealing, there is a fundamental conceptual inconsistency in policies that criminalise clients and purport to prioritise the safety of sex workers. In its original incarnation, the model of criminalising clients in Sweden was not designed to increase the safety of women in sex work; rather its goal was to eradicate prostitution and increase the safety of women who exit sex work. Indeed, in Sweden, the government explicitly condoned the increased risks that marginalised sex workers were exposed to by arguing that any adverse effects on women who remain in sex work were outweighed by the message of the law that prostitution is not tolerated.[28] Our findings indicate that policies that criminalise clients are, in practice, not reconciled with policies that aim to prioritise the safety of sex workers, such as outlined in VPD sex work enforcement guidelines. Indeed, the findings of our study indicate that despite police efforts to prioritise the safety of sex workers, when clients remain enforcement targets, sex workers continue to be at increased risk for physical and sexual violence and perceive police concern for their safety as a form of nuisance and harassment.

Street-involved sex workers, those living in poverty, Aboriginal sex workers and transgender sex workers have historically been exposed most directly to the negative effects of restrictive policing and criminalised sex work laws.[14 15 28] Evidence from Sweden indicates that criminalising clients negatively affected the working conditions and safety of all segments of sex workers by further pushing them underground. However, marginalised sex workers who solicit and/or service on the street were most negatively impacted by the criminalisation of clients, as they may not have the resources to reduce police scrutiny by advertising online, or be contacted by phone.[28] Owing to a less-developed welfare system in Canada, and the larger population of street-based sex workers compared with Sweden, the negative impact of demand criminalisation on sex workers' health and safety will likely be even more pronounced in the Canadian context.

The very existence of specific laws to regulate sex work speaks of the stigma associated with sex work and the link between sex work legislation and morality.[39] In Canada, as in most settings globally, there are already

laws in place for targeting various forms of exploitation and nuisance that may arise in the context of sex work, such as public disturbance, indecent exhibition, coercion, sexual assault, trafficking persons, extortion and kidnapping. As such, given the negative impact of criminalised sex work laws and enforcement practices, our findings lend further support to calls for the full decriminalisation of sex work in Canada, consistent with international guidelines by global policy bodies.[1 29]

New Zealand and parts of Australia have decriminalised sex work. In New Zealand, workplace health and safety standards have been established in consultation with sex workers, and sex workers can bring employment complaints to governing bodies.[27] The New Zealand Prostitution Reform Act enacted in 2003 treats sex work as any other business, regulating its commercial practice through standard employment health and safety regulations, regulating the location of commercial sex establishments through zoning by-laws and specifying the health and safety obligations of managers and workers.[39] Although decriminalisation is by no means a panacea, there is significant evidence to suggest that in New Zealand it has created improved working conditions for sex workers, including increased ability to report violence to police.[40] The removal of criminalised and enforcement-based approaches to sex work will also help to support access and implementation of other health and social interventions not currently available to the most criminalised street-involved sex workers, such as safer indoor work and housing spaces, integrated health and social welfare supports, substance-use prevention and treatment and opportunities for alternative employment for those sex workers who choose to transition out of sex work.

This study has limitations. The lived experiences of participants represented in this study reflect street-involved sex workers living in poverty and may not be representative of the experiences of sex workers in other segments of the industry. However, given that evidence has consistently shown that criminalisation and policing disproportionately target street-based sex workers, we feel these narratives provide critical evidence of the health and safety harms of such a policy.

In summary, this study suggests that enforcement strategies that target sex workers' clients reproduce the harms related to criminalised and quasi-criminalised approaches to the regulation of sex work. This empirical research clearly demonstrates that continued criminalisation and policing of clients, even in a context where sex workers no longer represent police targets, did not reduce the prevalence of sex work-related violence and profoundly impacted sex workers' ability to negotiate their working conditions, health and safety, including protections from violence, abuse and HIV/STIs. Sex workers were displaced to isolated areas with few protections from violence and abuse, were forced to rush or forgo screening clients and negotiating the terms of transactions (eg, fee and condom use) and agree to clients or sexual services they would otherwise refuse and

were unable to access police protections. The results also highlight the critical role of sex workers' lived experiences in any evidence-based policy making in Canada and globally. Two decades of failures of prostitution laws in Canada stemmed from failure of consecutive governments to listen to strong evidence by sex workers, academics and human rights experts that the laws were creating and exacerbating devastating harms to sex workers' safety, health and human rights, including violence, abuse and murder. In sum, this study raises serious questions about legislative approaches that criminalise clients and suggests that 'demand criminalisation' risks reproducing the devastating harms to health safety and human rights created by the criminalisation of sex work.

**Author affiliations**
[1]Gender and Sexual Health Initiative, British Columbia Centre for Excellence in HIV/AIDS, St. Paul's Hospital, Vancouver, British Columbia, Canada
[2]Pivot Legal Society, Vancouver, British Columbia, Canada
[3]Sex Workers United Against Violence, Vancouver, British Columbia, Canada
[4]Department of Medicine, University of British Columbia, Vancouver, British Columbia, Canada
[5]Urban Health Research Initiative, BC Centre for Excellence in HIV/AIDS, St. Paul's Hospital, Vancouver, British Columbia

**Acknowledgements** The authors would like to thank all the women who contributed their time and expertise to this project, as well as research, community partners and advisory board members. They wish to acknowledge Peter Vann, Gina Willis and Jennifer Morris for their research and administrative support.

**Contributors** KS is the principal investigator and senior author of the study and takes full responsibility for the integrity of the study procedures and data collection, management and analysis. AK, KP, JC and KS conceptualised this study and AK oversaw the field team carrying out the study. AK and CT conducted the interviews. AK and JC conducted the ethnographic observation. AK analysed the data using Atlas software, wrote the original draft of the article and incorporated feedback from all coauthors. KP, CT, JC, LB, SA, DB, TK, JSM and KS provided content expertise and critical feedback on the analyses and interpretation. All authors read and approved the final manuscript.

**Funding** This research was supported by operating grants from the US National Institutes of Health (R01DA033147 and R01DA028648).

**Competing interests** KS is partially supported by US National Institutes of Health (R01DA028648), Canadian Institutes of Health Research and Michael Smith Foundation for Health Research. JSM is supported by the British Columbia Ministry of Health and by the US National Institutes of Health (R01DA036307).

**Ethics approval** Providence Healthcare/University of British of Colombia Research Ethics Board.

**Provenance and peer review** Not commissioned; externally peer reviewed.

**Data sharing statement** No additional data are available.

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
