## [Reviewer comments · BMJ Open]

Some articles will have been accepted based in part or entirely on reviews undertaken for other BMJ Group journals. These will be reproduced where possible.

ARTICLE DETAILS

TITLE (PROVISIONAL)	CRIMINALISATION OF CLIENTS: REPRODUCING VULNERABILITIES FOR VIOLENCE AND POOR HEALTH AMONG STREET-BASED SEX WORKERS IN CANADA A QUALITATIVE STUDY
AUTHORS	Krüsi, Andrea; Pacey, Katrina; Bird, Lorna; Taylor, Christina; Chettiar, Jill; Allan, Sarah; Bennett, Darcie; Montaner, Julio; Kerr, Thomas; Shannon, Kate

VERSION 1 – REVIEW

REVIEWER	Gillian Abel University of Otago, New Zealand
REVIEW RETURNED	08-Apr-2014

GENERAL COMMENTS	I have a concern regarding a stated limitation on p27 line 34-36. The authors state that because the study is qualitative and therefore results are not generalisable, this is a weakness. Qualitative research does not aim to generalise so therefore why is this a weakness? What about the strengths qualitative research brings that generalisable research does not? If the research questions require a more in-depth investigation, generalisability is not an aim and therefore the lack of it is not a weakness. I think that the content of the paper is good and this is a very good opportunity to do this research in Canada with the transition in legislative environment. The more research carried out before a decision is made by government regarding how to regulate sex work the better. I did feel however, that some of the wording and sentence structure was clumsy and the paper could do with good editorial revision.
---

REVIEWER	Alexandra Lutnick RTI International, United States University of California, Berkeley, United States
REVIEW RETURNED	24-Apr-2014

GENERAL COMMENTS	With more countries moving towards "End Demand" approaches to sex work, this article is an extremely important addition to the literature. Through extensive qualitative/ethnographic work the authors reveal how increased policing of clients negatively impact sex workers. All of the interviews and observations focused on women (trans- and cisgender). In both the abstract and results section the authors note that the sample was comprised of "26 female and 5 transgender"
---

	sex workers. I recommend using the more recently favored terminology of cisgender and transgender women. While reviewing the article I found myself wondering if the analysis revealed thematic differences based on gender, race, or primary way of solicitation. In the manuscript's current state it appears that regardless of these differences, all the women shared a similar experience with increased policing of clients. If this is not true, please better highlight this in the findings and discussion. I recognize the importance of obscuring the identity of respondents and why the authors may have favored identifying respondents as "participant #". Similar to my previous comment I think the qualitative data would be enhanced if it could be identified in a more personal way. Perhaps the authors could include some combination of gender identity, age, ethnicity, and solicitation and transaction venues? The inclusion on the prostitution-related criminal code offenses is a nice addition to the data. Not all readers will be familiar with the ways in which sex workers are charged with laws that one typically thinks would be confined to clients or facilitators. It may be helpful to briefly describe this. I also wonder if any of the qualitative data highlighted how sex workers were charged with non-prostitution related offenses? I have attached a pdf of the manuscript with some very minor editorial suggestions.
--	--

VERSION 1 – AUTHOR RESPONSE

Reviewer 1:

Reviewer Name: Gillian Abel

Institution and Country: University of Otago, New Zealand

1. I have a concern regarding a stated limitation on p27 line 34-36. The authors state that because the study is qualitative and therefore results are not generalisable, this is a weakness. Qualitative research does not aim to generalise so therefore why is this a weakness? What about the strengths qualitative research brings that generalisable research does not? If the research questions require a more in-depth investigation, generalisability is not an aim and therefore the lack of it is not a weakness.

Author Response: We fully agree with Reviewer 1 regarding the strength and weaknesses of a qualitative research approaches and have therefore removed the statement regarding the generalizability of this work from the limitations in the discussion section. We have also highlighted the importance of qualitative research in bringing forward the narratives and lived experience of sex workers.

2. I think that the content of the paper is good and this is a very good opportunity to do this research in Canada with the transition in legislative environment. The more research carried out before a decision is made by government regarding how to regulate sex work the better.

Author Response: We agree and appreciate the quick-turn around of reviewers and editors to ensure this paper is able to inform current policy debates in Canada before new legislation is introduced, likely later this month.

3. I did feel however, that some of the wording and sentence structure was clumsy and the paper could do with good editorial revision.

Author Response: We agree and have thoroughly reviewed the paper for 'readability' and have made several edits to increase the flow of the manuscript.

Reviewer 2:

Reviewer: 2

Reviewer Name: Alexandra Lutnick

Institution and Country: RTI International, United States

University of California, Berkeley, United States

1. With more countries moving towards "End Demand" approaches to sex work, this article is an extremely important addition to the literature. Through extensive qualitative/ethnographic work the authors reveal how increased policing of clients negatively impact sex workers.

Author Response: We completely agree with the Reviewer that this research on the experiences of sex workers is highly timely and relevant to global policy debates around "End Demand" approaches in many countries.

2. All of the interviews and observations focused on women (trans- and cisgender). In both the abstract and results section the authors note that the sample was comprised of "26 female and 5 transgender" sex workers. I recommend using the more recently favored terminology of cisgender and transgender women.

Author Response: We completely agree with the Reviewer 2's suggestion regarding the use of cis- and transgender terminology and have adjusted the text throughout to address this key point (see both abstract and results below).

Abstract: "Participants: 26 cisgender and 5 transgender women street-based sex workers (n=31) participated in semi-structured interviews about their working conditions."

Results: "The sample for semi-structured interviews included 26 cisgender and 5 transgender women sex workers (total n=31)."

3. While reviewing the article I found myself wondering if the analysis revealed thematic differences based on gender, race, or primary way of solicitation. In the manuscript's current state it appears that regardless of these differences, all the women shared a similar experience with increased policing of clients. If this is not true, please better highlight this in the findings and discussion.

Author Response: We fully agree with Reviewer 2 that the manuscript is improved by further contextualising the qualitative interview data that is presented. We have highlighted in the text that across the different demographics of street-based sex workers the impact of police enforcement of sex work clients were experienced similarly without pronounced distinctions according to gender, ethnicity or primary way of solicitation. While previous research both by our group and others in this setting (and elsewhere) has shown differences in policing experiences for sex workers shaped by ethnicity and gender, we did not see these in our current results. We have added the following text to the results section:

Sex Workers' Experiences with New Sex Work Enforcement Guidelines

"The vast majority of sex workers, regardless of gender, ethnicity and primary place of solicitation, reported that their interactions with police when soliciting sex work clients are more positive and

generally focus on their safety.“

4. I recognize the importance of obscuring the identity of respondents and why the authors may have favored identifying respondents as "participant #". Similar to my previous comment I think the qualitative data would be enhanced if it could be identified in a more personal way. Perhaps the authors could include some combination of gender identity, age, ethnicity, and solicitation and transaction venues?

Author Response: We agree that bring forward the qualitative data of sex workers in a most personal way possible is important, and we have removed "participant ID" and instead opted for the pseudonyms assumed by women during the interviews to protect anonymity as well as gender (e.g. "Fiona, Transgender Women Sex Worker"). We have also added "sex workers voices" to each of the tables/panels.

5. The inclusion on the prostitution-related criminal code offenses is a nice addition to the data. Not all readers will be familiar with the ways in which sex workers are charged with laws that one typically thinks would be confined to clients or facilitators. It may be helpful to briefly describe this. I also wonder if any of the qualitative data highlighted how sex workers were charged with non-prostitution related offenses?

Author Response: We thank the Reviewer for this request for clarification. We have outlined the three laws as they relate to sex work, under which sex workers or clients could be charged. As noted, while unfortunately police statistics do not separate charges by workers or clients, the VPD policy and public communication by police indicates that police have not been arresting sex workers under prostitution-related offenses since the implementation of the 2013 policy, so one can assume these were arrests of clients. The current qualitative data did reveal how sex workers and clients are now pulled over for other offenses (e.g. trafficking violations) and then once pulled over, police may charge clients, and let the sex workers go. We have added text to this effect in the results and discussion. This continues to create distrust with police and increases need to avoid police scrutiny thereby displacing sex work. We also agree that beyond these examples, further exploration of other non-prostitution related charges (e.g. administrative offenses such as public nuisance) will be important to investigate going forward. We added the following text:

Results:

Continued Police Enforcement of Sex Buyers (Clients)

"In some cases, police used non-prostitution related charges (e.g. trafficking violations) as a means to pull over the vehicle and target clients."

Discussion:

"Sex workers also described how police used other non-prostitution related offenses (e.g. administrative laws such as trafficking violations, public nuisance) to target sex workers and clients."

6. I have attached a pdf of the manuscript with some very minor editorial suggestions.

Author Response: We would also like to thank Reviewer 2 for the editorial suggestions. We have incorporated them.